Full-length transcriptome profiling of Gentiana straminea Maxim. provides new insights into iridoid biosynthesis pathway

Yang Lina 1
He Tao hetaoxn@aliyun.com 2 3
Wang Le wangleqhu@163.com 3
Ning Xiaochun 4
Wang Shuai 3
1 College of Agriculture and Animal Husbandry, Qinghai University , Xi’ning , Qinghai , China
2 School of Ecol-Environmental Engineering, Qinghai University , Xi’ning , Qinghai , China
3 State Key Laboratory of Plateau Ecology and Agriculture, Qinghai University , Xi’ning , Qinghai , China
4 Xining Center of Natural Resources Comprehensive Survey, China Geological Survey , Xi’ning , Qinghai , China
Uversky Vladimir
Electronic publication date: 2025 Oct 23
Publication date: 2025
Volume: 13
Electronic Location ID: e20136
Received 2025 Feb 5; Accepted 2025 Sep 4
Copyright: ©2025 Yang et al.
Copyright year: 2025
Copyright holder: Yang et al.
License: This is an open access article distributed under the terms of the Creative Commons Attribution License, which permits unrestricted use, distribution, reproduction and adaptation in any medium and for any purpose provided that it is properly attributed. For attribution, the original author(s), title, publication source (PeerJ) and either DOI or URL of the article must be cited.
License URL: https://creativecommons.org/licenses/by/4.0/

Keywords: Gentiana straminea, SMRT sequencing, Iridoid biosynthesis, Expression profile

Funding: The Youth Fund projects, Qinghai University No. 2020-DNY-7 Central Leading Local Science and Technology Development Fund Project 2024ZY030 This research was supported by the Youth Fund projects, Qinghai University (No. 2020-DNY-7) and Central Leading Local Science and Technology Development Fund Project (2024ZY030). The funders had no role in study design, data collection and analysis, decision to publish, or preparation of the manuscript.

==============================
Gentiana straminea Maxim. is a traditional Chinese medicinal plant celebrated for its diverse array of bioactive compounds, particularly iridoid glycosides. These compounds are recognized as the main components that exert therapeutic effects against rheumatism, osteoarthritis, hepatitis, gastritis, and cholecystitis. Consequently, research on G. straminea has attracted significant interest, yet the genetic factors underlying the production and diversification of its secondary metabolites remain poorly understood, especially the pathway of iridoid biosynthesis. In the present study, a full-length transcriptome analysis based on PacBio Sequel II platform and Illumina sequencing was performed to identify genes that were differentially expressed in five G. straminea tissues, and proteins catalyzing iridoid biosynthesis were characterized. After sequence clustering and redundancy removal, a total of 32,776 isoforms were identified in PacBio sequencing, with an average length of 2,589.14 bp, an N50 value of 2,767 bp, and a guanine-cytosine (GC) content of 41.43%. Results of Illumina sequencing unveiled that a total of 31,330 genes were found in common in all the five tissues. Kyoto Encyclopedia of Genes and Genomes (KEGG) enrichment analysis revealed that the DEGs were mainly enriched in terms related to biosynthesis of secondary metabolites, metabolic pathways, mitogen-activated protein kinase (MAPK) signaling pathway, etc. A total of 117 isoforms encoding 19 key enzymes related to the iridoid synthesis pathway were identified, including one geranyl diphosphate synthase (GPPS) and two geranylgeranyl diphosphate synthases (GGPPS). A phylogenetic analysis further classified plant G(G)PPSs into three distinct branches. Profiling tissue-specific expression of key genes involved in iridoid synthesis revealed that the Quantitative reverse transcription polymerase chain reaction (RT-qPCR) results demonstrated the consistent trend with the FPKM (Fragments Per Kilobase of transcript per Million mapped reads) values in the root, stem, leaf, flower, ovary, non-embryonic callus (NEC), and embryonic callus (EC). Among them, AACT, IDI, ISPH, and GCPE had the highest expression levels in leaves, whereas DXS and GPPS had the highest expression levels in stems. This work provides the first transcriptomic analysis of G. straminea, serving as a foundational resource for elucidating mechanisms of bioactive compound biosynthesis, facilitating molecular studies, and supporting genomic exploration of this medicinal species.

Introduction

Gentiana straminea Maxim., a member of the Gentianaceae family, is a medicinal plant widely used in traditional Chinese medicine (Ye et al., 2021). This species is primarily distributed in high-altitude regions (2,000–4,950 m) of Qinghai, Xizang and Sichuan, where it thrives in alpine meadows, forests, and grasslands (Jia et al., 2012). Previous studies have indicated that iridoids from the roots of G. straminea have therapeutic effects against rheumatism, osteoarthritis, hepatitis, gastritis, and cholecystitis (Zhou et al., 2016). The main medicinal effects are associated with gentiopicroside, loganic acid, sweroside and swertiamarin, which are all iridoid compounds (Wei et al., 2012; Wu et al., 2016). Because of their abundant pharmacological properties, iridoid have emerged as a research hotspot in related studies. To date, the research on the iridoid biosynthetic pathway is relatively well-established primarily in Catharanthus roseus (Oudin et al., 2007). The biosynthesis pathway of iridoids in G. straminea remains poorly understood. Elucidating its biosynthesis mechanism has become essential for targeted enhancement of pharmacologically active compounds in G. straminea.

Preliminary results demonstrate that iridoids (oxygenated monoterpene structurally derived from two isoprene units), are biosynthesized via a three-step biosynthetic pathway. The first stage involves synthesis of the precursors isopentyl diphosphate (IPP) and dimethylallyl diphosphate (DMAPP), which can be formed via the mevalonate pathway (MVA) and the methylerythritol phosphate pathway (MEP). MVA occurs mainly in the cytoplasm, whereas MEP occurs mainly in plastids (Zhan et al., 2023). The second stage involves the formation of the iridoid skeleton, during which IPP and DMAPP are catalytically condensed by geranyl diphosphate synthases (GPPS) to produce geranyl diphosphate (GPP), while geranylgeranyl diphosphate (GGPP) is generated through a catalytic process mediated by geranylgeranyl diphosphate synthases (GGPPS) (Vranová, Coman & Gruissem, 2013). Then, GPP and GGPP are used as the raw materials for the synthesis of different terpenoids (monoterpenes, diterpenes, triterpenes, etc.) through different metabolic pathways (Sun et al., 2012). The third stage is the synthesis of iridoid from GPP. GPP is converted to geraniol by geranyl diphosphate diphosphatase (GES)-mediated catalysis and hydrolysis (Oudin et al., 2007). Geraniol undergoes structural modifications including glycosylation, hydroxylation, methylation, and isomerization, along with other reactions, to yield iridoid derivatives (Zhao & Wang, 2020). Gentiopicroside, as one of these derivatives, possesses significant medicinal value in Gentianaceae plants (Wu et al., 2016).

Advances in sequencing technology have facilitated the application of transcriptome sequencing in transcript profiling, gene discovery, and determining which genes are expressed in plants. The transcriptome consists of all the RNA transcripts of a species and reflects the functions of different cells and tissues during a particular period. Modern high-throughput RNA sequencing technologies enable the analysis of genes that regulate the synthesis of secondary metabolites in non-model species. This approach can uncover new genes, potential metabolic pathways, and associated genetic regulatory mechanisms (Ozsolak & Milos, 2012). The third-generation single-molecule real-time sequencing (SMRT) enables sequencing of transcripts up to 10 kb without a reference genome. However, it is limited by high cost per base, high error rates, and low throughput (Rhoads & Au, 2015). Second-generation sequencing produces short read lengths, but provides high sequencing accuracy. Due to limitations imposed by its read length and assembly algorithms, second-generation sequencing cannot accurately obtain the complete sequence of transcripts, particularly for different transcripts with high homology. Consequently, the synergistic integration of second- and third-generation sequencing techniques enables the generation of high-fidelity sequencing data with low error rate. Full-length transcriptome-based Illumina sequencing has been applied in research involving Coptis deltoidei (Zhong et al., 2020), Ranunculus japonicus (Xu et al., 2023), Fritillaria hupehensis (Guo et al., 2021), Angelica sinensis (Gao et al., 2021), Torreya grandis (Lou et al., 2019), and Salvia miltiorrhiza (Xu et al., 2016).

This study aimed to identify and characterize key genes involved in iridoid biosynthesis in G. straminea through an integrated multi-omics approach. Combining third-generation (PacBio) and second-generation (Illumina) sequencing platforms, we obtained full-length transcriptomes and quantified tissue-specific expression patterns of iridoid pathway-related genes. Candidate genes were further validated via RT-qPCR, while differentially expressed genes (DEGs) across tissues were functionally annotated using KEGG enrichment analysis. These findings establish a critical genetic foundation for deciphering the iridoid biosynthetic pathway in G. straminea and will facilitate future metabolic engineering efforts.

Materials & Methods

Preparation and collection of samples for transcriptome sequencing and qPCR analysis

Samples of G. straminea individuals were collected in August 2023 during the flowering stage from Yushu, Maqin County, Qinghai Province, China (N34°38′380, E100°23′546, altitude 4,200 m). Fresh tissues from five types—root, stem, leaves, flower, and ovary—were collected, washed with sterilized water, wrapped in foil, and then preserved in liquid nitrogen. Tissues of these five types were further utilized for library construction and SMRT sequencing. Additionally, for the quantification of gene expression via qPCR, two additional tissue types—non-embryonic callus (NEC) and embryonic callus (EC)—were included. The generation of EC and NEC tissues was carried out following the protocol outlined by He, Yang & Zhao (2011), utilizing leaves as explants. Three biological replicates were collected for all samples.

RNA extraction and SMRT sequencing

Samples of G. straminea were used for total RNA extraction on ice, according to the manufacturer’s protocol, with TRIzol reagent (Life Technologies, Karlsbad, CA, USA). An Agilent 2100 Bioanalyzer and agarose gel electrophoresis were used to determine the integrity of the total RNA. A Nanodrop microspectrophotometer (Thermo Fisher Scientific, Waltham, MA, USA) was used to check the purity and concentration of the RNA. The Clontech SMARTer PCR cDNA Synthesis Kit was used to reverse transcribe the oligo (dT) magnetic bead-enriched mRNA to cDNA. PCR cycle optimization was employed to identify the ideal number of amplification cycles for subsequent large-scale PCRs. Double-stranded cDNAs were generated with the optimized cycle number. Additionally, size selection at >5 kb and equal mixing without size selection of cDNA were performed with the BluePippinTM Size Selection System. The next step in the construction of the SMRTbell library was carried out by large-scale PCR. The sequencing primer was matched with the SMRTbell template by annealing, and then linked to the polymerase. Sequencing was conducted using the PacBio Sequel II platform at Gene Denovo Biotechnology Co.

The raw sequencing reads from the cDNA library were classified via the Pacific Biosciences Iso-Seq pipeline, with high-quality circular consensus sequence (CCS) first extracted. Transcript integration was assessed according to whether the CCS reads contained all 5′primers, the 3′primer and the poly-A sequences. Full length sequences (FLs) were those that contained all three sequences. After the removal of primers, barcodes and poly A tails, full-length nonchimeric (FLNC) reads were obtained. Reads less than 50 bp in length were discarded. The entire isoform was generated by clustering the FLNC reads. Minimap2 was used for similar FLNC reads, which were then clustered hierarchically to obtain a consistency sequence (unpolished consensus isoforms). The consistency sequence was then further corrected via the Quiver algorithm. The high-quality isoforms (prediction accuracy ≥ 0.99) were used for subsequent analysis.

Library construction and Illumina sequencing

Total RNA was enriched by Oligo(dT) beads to form mRNA, then was fragmented into short fragments. With random primers, fragments were transcribed into cDNA, then syntheisized the second-strand cDNA with DNA polymerase I, Rnase H, dNTP and buffer. the obtained cDNA was purified with QiaQuick PCR extraction kit (Qiagen, Venlo, The Netherlands), end repaired, poly(A) added, and ligated to Illumina sequencing adapters. The ligated products were screened by agarose gel electrophoresis, amplified by PCR, and sequenced by Gente Denovo Biotechnology Co. (Guangzhou, China) using Illumina HiSeqTM 4000. High quality clean reads were obtained by fastp (Version 0.18.0), with removing adapters, containing more than 10% of unknown nucleotides (N) and low-quality reads.

Isoform expression and differential expression analysis

Using the full-length transcriptome as the reference, the clean and high-quality reads were mapped using RSEM (version 1.2.8) to determine the isoform expression in five different tissues of G. straminea. The expression levels of isoforms from each sample were calculated and normalized to FPKM. Differentially expressed genes (DEGs) were identified using the DESeq2 software with —log2 (Fold Change)— ≥ 2 and a false discovery rate (FDR) below 0.05 as the screening criteria.

Functional annotation, structure analysis

The sequences of the isoforms were checked against the non-redundant protein (Nr) database of the NCBI (http://www.ncbi.nlm.nih.gov), the Clusters of Orthologous Genes/EukaryoticOrthologous Group (COG/KOG) database (http://www.ncbi.nlm.nih.gov/COG), the Kyoto Encyclopedia of Genes and Genomes (KEGG) database (http://www.genome.jp/kegg), and the Swiss-Prot protein database (http://www.expasy.ch/sprot) via the BLASTx program (http://www.ncbi.nlm.nih.gov/BLAST/), with an E value threshold of 1e−5, to assess the similarity of the sequences to those of genes from other species. Gene Ontology (GO) annotation was analyzed using isoforms from the Nr annotation results by Blast2GO software. The top 20 scoring isoforms and no fewer than 33 high-scoring segment pair hits (HSPs) were selected for the Blast2GO analysis. Isoforms were functionally classified using WEGO software. Transcription factors (TFs) were predicted via hmmscan by aligning the protein coding sequences to the Plant TFdb (https://planttfdb.gao-lab.org/).

Identification of the putative iriodoid biosynthesis pathway

Based on previous literature and known information from public databases, candidate isoforms for iridoid biosynthesis were screened from the sequence annotation files derived from third-generation sequencing. These candidate isoforms were subsequently annotated against the Nr, Swiss-Prot, KEGG, and KOG databases. Isoforms annotated with Enzyme Commission numbers (EC numbers) and keywords associated with terpenoid backbone biosynthesis, cytochrome P450, and iridoids were selected for further analysis. The screened isoforms were mapped to reference pathways, such as terpenoid backbone biosynthesis (map000900) and monoterpenoid biosynthesis (map00902), within the KEGG database. With reference to published core iridoid biosynthesis pathways (Oudin et al., 2007; Miettinen et al., 2014; Ni et al., 2019; Liu et al., 2017; Salim et al., 2014; Rai et al., 2013; Xu et al., 2022). These putative enzyme isoforms were assigned to corresponding biochemical reaction steps, thereby establishing a putative pathway for iridoid biosynthesis in G. straminea.

Protein-protein interaction network analysis

Functional enrichment analysis of protein-protein interaction (PPI) networks for genes involved in the iridoid biosynthesis pathway was conducted using the String database (https://cn.string-db.org/) (Szklarczyk et al., 2014). The protein sequences corresponding to key enzymatic components of the iridoid biosynthetic machinery in G. straminea were queried against the string database, with A. thaliana designated as the reference model organism for orthologous mapping. After sequence homology analysis, the highest-confidence protein isoforms (with open reading frames, ORFs) were selected based on maximal sequence identity from the mapping interface. Network construction parameters were optimized as follows: Minimum required interaction score: High confidence (score ≥ 0.700); max number of interactors to show: first shell limited to query proteins only; visualization output: dynamic Scalable Vector Graphics (SVG) network representation.

Identification and bioinformatic analysis of G(G)PPSs in G. straminea

For identification of GsG(G)PPSs, local BLAST search was performed using GGPPSs from Arabidopsis thaliana (Beck et al., 2013) or Chimonanthus praecox (Kamran et al., 2020) as queries. A threshold of e-value <10−10 was applied for preliminary screening. After searching, sequences of putative GsG(G)PPSs were further subjected to CDD (https://www.ncbi.nlm.nih.gov/cdd/) and InterPro (https://www.ebi.ac.uk/interpro/result/InterProScan/) for domain confirmation (Paysan-Lafosse et al., 2023). Prediction and analysis of the physicochemical properties of the GsG(G)PPS amino acid sequences were performed via ExPASy (https://web.expasy.org/protparam/) (Artimo et al., 2012). Sequences were submitted to SignalP4.1 server for prediction of the signal peptide (https://services.healthtech.dtu.dk/services/SignalP-4.1/) (Petersen et al., 2011). Subcellular localization was determined via WoLF PSORT (https://wolfpsort.hgc.jp/), transmembrane structure was predicted by HMHMM2.0 (https://services.healthtech.dtu.dk/services/TMHMM-2.0/). In addition, the annotated sequence information was submitted to the MEME website (https://meme-suite.org/meme/doc/meme.html), using 6–100 residues as the optimal motif size to search for 10 conserved motifs and predicted the conserved protein motifs in the sequence (Bailey et al., 2015). Similar to the GsGGPPS SSU, GsGGPPS, and GsGPPS amino acid sequences were downloaded from NCBI BLAST (https://blast.ncbi.nlm.nih.gov/Blast.cgi). Protein sequence alignment was performed via DNAMAN. Protein structure prediction was performed via SWISS-MODEL (https://swissmodel.expasy.org/).

Phylogentic analysis of G(G)PPS gene family

The GsG(G)PPSs obtained, and G(G)PPSs from other species were incorporated for phylogenetic analysis. G(G)PPSs in Arabidopsis thaliana and Nicotiana tabacum genomes were identified through BLAST by using the ensemble database (https://asia.ensembl.org/Multi/Tools/Blast). G(G)PPS homologues from other species available in the NCBI database were included for phylogenetic analyses of the G(G)PPS family. Details on all the G(G)PPSs used for phylogenetic analysis were listed in Table S1. These species protein sequences were performed multiple sequence alignment using the ClustalW program integrated in MEGA11.0. Phylogenetic inference of G(G)PPSs was conducted using the neighbor-joining method in MEGA 11.0 software, with a bootstrap test of 1,000 replicates (Tamura, Stecher & Kumar, 2021) and the best-fit substitution model JTT+G+I. The refinement of the evolutionary tree was completed using the online software Evoview (https://www.evolgenius.info/evolview/#/).

Expression analysis of key enzymes by real—time quantitative PCR

First-strand cDNA synthesis was performed using a cDNA reverse transcription kit (PrimeScriptTMII 1st Strand cDNA Synthesis Kit), following the protocol provided. Primers for reverse transcription-quantitative polymerase chain reaction (RT-qPCR) were designed using the OligoArchitect online sever and synthesized by Sangon Biotech (Shanghai) Co., Ltd. The primers sequence shown in Table S2, qPCR was performed using TB Green Premix Ex Taq with a 20 μL reaction system, which included 10 μL of TB Green Premix, 0.8 μL each of forward and reverse primers (10 μM), two μL of cDNA, 6.4 μL of ddH2O. The reaction procedure consisted of the following steps: pre-denaturation at 94 °C for 5 min, denaturation at 94 °C for 30 s, annealing at 53 °C for 30 s, extension at 72 °C for 30 s, followed by 40 cycles. The GAPDH gene was utilized as the internal reference for relative expression analysis. The quantification of gene expressions was conducted using three biological replicates. Relative expression was calculated using the Ct (2−ΔΔCt) method, following the approach described by Livak & Schmittgen (2001), with root expression serving as the control. The significance analysis of difference tissue was conducted with means of gene expression by the Duncan test at 5%.

Results

Transcriptome sequencing of G. straminea

Both SMRT and Illumina sequencing were performed on the root, stem, leaf, flower, and ovary tissues of G. straminea. The average amount of raw data generated was 6.5 GB for second-generation sequencing per sample (Table 1). For PacBio sequencing, a total of 62.47 GB of raw data was obtained. The per-base coverage depth reached approximately 45× (circular consensus sequences corrected depth). A total of 23,318,162 subreads were generated from third-generation sequencing. After self-correction and merging, 499,496 circular consensus sequences (CCS) were generated, with an average CCS read length of 2,789 bp. The distributions for the number of passes and read lengths of these CCS reads are shown in Figs. S1A–S1B. The full-length nonchimeric sequences with high-precision CCS reads were identified, and similar FLNC reads were clustered hierarchically to obtain consensus sequences (Fig. S1C). A total of 41,785 high-quality isoforms (HQs) and 140 low-quality isoforms (LQs) were obtained after further correction. After removing redundant sequences, the total length of the isoforms was 84,861,577 bp. A total of 32,776 isoforms were obtained, and the lengths ranged from 165 to 10,169 bp, with an average length of 2,589.14 bp, an N50 of 2,767 bp, and a guanine-cytosine (GC) content of 41.43%. The number and length distribution of the isoforms are shown in Fig. S1D.

Functional annotation of the full-length transcriptome of G. straminea

The HQ unigenes were annotated against four functional annotation databases Nr, Swiss-Prot, KEGG, and KOG. A total of 31,434 (95.9%) unigenes were successfully annotated, while 1,342 were unannotated. The highest number of unigenes (31,235; 97.47%) was annotated to Nr database, followed by the KEGG database and the Swiss-Prot database, in which 30,990 (94.55%) and 27,622 (84.27%) unigenes, respectively, were annotated. The lowest number of unigenes was annotated in the KOG database (22,753; 69.42%). Summary, 21,742 common unigenes (66.34%) were annotated in all four databases (Fig. 1A). These findings were compared with those for 414 species annotated in the Nr database (top ten shown in Fig. 1B). The species with the most annotated sequence information was Coffea arabica, with 8,701 (27.86%) unigenes, followed by Coffea eugenioides, Coffea canephora, and Olea europaea, with 5,318 (17.03%), 3,512 (11.24%), and 1,051 (3.36%) unigenes, respectively.

Table 1 Comparison with reference gene sequence Pure reads obtained in second generation sequencing.

Sample	CleanData (GB)	Total_Mapped (%)	Unique_Mapped (%)	
R-1	6.574	76.63	18.14	
R-2	6.944	76.75	18.04	
R-3	6.449	76.95	18.08	
S-1	6.712	73.75	18.86	
S-2	5.728	73.86	18.9	
S-3	5.835	73.33	19.04	
L-1	6.398	76.13	20.3	
L-2	5.922	76.74	19.34	
L-3	6.232	76.95	19.39	
F-1	6.507	71.36	18.76	
F-2	6.837	70.92	18.59	
F-3	6.406	71.18	18.65	
O-1	7.384	72.48	19.04	
O-2	6.594	71.91	18.92	
O-3	7.001	72.3	18.89	
Notes.

R tissue of root

S stem

L leaf

F flower

O ovary; the numbers after letters indicated three biological replicates

Figure 1 Venn diagram and species distribution.

(A) Venn diagram showing the number of unigenes annotated to four databases; (B) the top ten species distribution annotated in the Nr database.

The KOG analysis identified 22,753 unigenes, which could be classified into 25 categories (Fig. 2). The largest number of annotated genes was associated with general function prediction only 4,733 genes (20.80%), followed by 4,146 genes (18.22%) annotated to signal transduction mechanisms; 2,859 genes (12.57%) annotated to posttranslational modifications, protein turnover, and chaperones; 1,617 genes (7.11%) annotated to carbohydrate transport and metabolism; and 1,533 genes (6.74%) annotated to RNA processing and modification. The lowest number was observed for cell motility (37; 0.16%). In addition, 1,210 (5.30%) genes with unknown functions were identified.

Figure 2 KOG function classification.

The unigenes annotated by GO function analysis were associated with 51 GO terms, which were grouped into three categories: cellular component, molecular function, and biological process (Fig. S2). The top three GO enriched terms in the biological process category were cellular process, metabolic process, and response to stimulus, with 21,052, 18,417 and 7,559 genes, respectively. The top three enriched GO terms in the molecular function category were binding (18,864), catalytic activity (16,549), and transporter activity (3,175). In the cellular component category, cellular anatomical entity (16,552) and protein-containing complex (6,946) terms were highly enriched.

In the KEGG database, 30,990 unigenes of G. straminea were annotated and classified into five main categories and 19 subclasses (shown in Table S3). Of these annotated unigenes, 9,485 were mapped to specific KEGG pathways. The greatest number of genes (4,647; 48.99%) was annotated in metabolism pathways, followed by secondary metabolite biosynthesis (2,494; 26.29%), carbon metabolism (826; 8.71%), and biosynthesis of amino acids (635; 6.69%) (Table S4). Genes annotated to secondary metabolite biosynthesis pathways may be related to the synthesis of the medicinal components of G. straminea. In addition to carbon metabolism, the biosynthesis of amino acids and other metabolic pathways may be related to cellular osmotic regulation and the oxidative stress response. These annotated genes provide important sequence information for investigating the biosynthetic mechanism of the metabolites of G. straminea.

In this study, 708 annotated genes were found to participate in 20 standard KEGG secondary metabolism pathways in the transcriptome of G. straminea (Table S5); among these genes, 121 genes were annotated to the terpenoid backbone biosynthesis pathway and 67 genes were enriched in terpenoids (monoterpenoid, diterpenoid, sesquiterpenoid and triterpenoid biosynthesis) pathways. Furthermore, there were 84 genes involved in phenylpropanoid biosynthesis, while 41 genes were involved in flavonoids, isoflavonoid, flavone and flavonol biosynthesis. Additionally, 89 genes related to the synthesis of various alkaloids (indole, isoquinoline, tropane, piperidine, and pyridine alkaloid biosynthesis) were annotated, as shown in Table S5.

Predicting TFs

According to the assembly results, 1,151 genes were annotated as TFs, distributed in 51 TF families. The largest number of genes belonged to the GRAS family, with 128 genes (11.12%), followed by the ARF, C3H, bHLH, and WRKY families, with 92, 70, 66 and 66 genes, respectively. The least common families were the NF-YB (1), M-type (1), Whirly (1), AP2 (1), and YABBY (1) families. The ten TF families with the greatest number of genes in G. straminea are shown in Fig. S3.

DEGs analysis

A total of 32,470 genes were detected, and venn diagram analysis revealed that 31,330 genes were commonly found in the five tissues (Fig. 3B). In the comparisons of root and stem, root and leaf, root and flower, root and ovary, a total of 9,809, 10,503, 13,195, 9,699 DEGs were identified, respectively. Among these, 6,594, 5,762, 6,572, 5,727 DEGs were up-regulated and 3,260, 4,741, 6,623, 3,972 DEGs were down-regulated, respectively. In the contrast between leaf and stem, leaf and flower, leaf and ovary, a total of 6,980, 10,475, 10,006 DEGs were separately detected. Of these, 4,030, 4,707, 5,002 DEGs were up-reaulated and 2,950, 5,768, 5,004 DEGs were down-regulated. Furthermore, 8,855, 7,021 DEGs were identified in the stem vs flower, stem vs ovary comparison groups, respectively. Comparing with ovary, in the tissue of flower, 3,456 DEGs were up-regulated, and 2,218 DEGs were down-regulated, as shown in Fig. 3A.

Figure 3 Distribution of DEG expression level across experimental groups and venn diagram of tissue-species DEGs.

(A) Up-regulated and down-regulated number distribution of DEG expression level across experimental groups. (B) Venn diagram of DEG in different tissues.

In the comparisons of five different tissues, DEGs annotated were mainly enriched in the metabolic pathways, biosynthesis of secondary metabolites. Additionally, DEGs genes were enriched in the pentose and glucuronate interconversions in the root-vs-flower and root-vs-ovaries; carbon metabolism in leaves-vs-stem and flowers-vs-ovaries; amino sugar and nucleotide sugar metabolism in roots-vs-stem (Fig. S4).

Analysis of iridoid biosynthesis genes in G. straminea

Iridoid compound, which are common secondary metabolite components found in various medicinal plants, are the main components of G. straminea and have significant biological activity. By combining these results with previous research results (Ni et al., 2019; Liu et al., 2017), we identified a putative pathway for iridoid biosynthesis and the isoforms involved (Fig. 4). Our results revealed that 117 isoforms encoded 19 key enzymes (Table S6). The expression levels of these key enzyme isoforms in seven tissues are shown with a heatmap (Fig. 4), of which, GCPE, STR, ISPE, DXR, ISPH,7-DLS showed relatively high expression in leaves, other genes showed different expression patterns in different tissues. To further elucidate the interactions among these genes. The STRING database was employed to construct a protein-protein interaction (PPI) network for proteins annotated in the iridoid biosynthesis pathway of G. straminea. A. thaliana was used as the reference organism, the annotated protein sequences were uploaded to STRING database. Initial analysis revealed only 18 associated proteins, with subsequent network formation showing just 15 functionally connected proteins (Fig. 5A). The resulting PPI network comprised 15 nodes and 82 edges (Fig. 5A), exhibiting an average node degree of 10.9 and an average local clustering coefficient of 0.863. The PPI enrichment p-value was highly significant (<1.0e−16). STRING database annotations indicated experimentally validated interactions between: GGPPS SSU and GGPPS (Purple edges), AACT and HMGCS (light blue edges). The other interacting pairs were derived from curated databases and predicted interactions. The protein interaction network-associated genes displayed distinct tissue-specific expression profiles (Fig. 5B).

Figure 4 Putative pathways and heatmap of isoforms related to iridoid biosynthesis in different tissues of G. straminea.

Note: Enzymes highlighted in red indicate annotated genes in G. straminea, while black labels denote unannotated homologs, the red numerals in parentheses represent the isoform count in G. straminea. Different color-coded blocks visualized tissue-specific expression patterns across seven tissues types (R, root; S, stem; L, leaf; F, flower; O, ovary; NEC, non-embryonic callus; EC, embryonic callus). With each row corresponds to a normalized expression profile (FPKM) of iridoid biosynthesis-related isoforms. The heatmap employs a row-wise-z-score normalization, where red indicates high relative expression, and blue indicates low relative expression. AACT, Acetyl-CoA C-acetyltransferase; HMGCS, Hydroxymethylglutaryl-CoA synthase; HMGCR, Hydroxymethylglutaryl-CoA reductase (NADPH); MVK, Mevalonate kinase; PMK: Phosphomevalonate kinase; MVD, Diphosphomevalonate decarboxylase; IDI, Isopentenyl-diphosphate delta-isomerase; DXS, 1-Deoxy-D-xylulose-5-phosphate synthase; DXR, 1-Deoxy-D-xylulose-5-phosphate reductoisomerase; ISPD, 2-C-methyl-D-erythritol 4-phosphate cytidylyltransferase; ISPE, 4-Diphosphocytidyl-2-C-methyl-D-erythritol kinase; ISPF, 2-C-methyl-D-erythritol 2,4-cyclodiphosphate synthase; GCPE, (E)-4-Hydroxy-3- methylbut-2-enyl-diphosphate synthase; ISPH, 4-Hydroxy-3-methylbut-2-en1yl diphosphate reductase; GPPS, Geranyl diphosphate synthase; GGPPS, Geranylgeranyl diphosphate synthase; GES, Geranyl diphosphate diphosphatase; POR, Cytochrome P450 reductase; G10H, Geraniol 10 -hydroxylase; 10-HG O, 10 -Hydroxygeraniol oxidoreductase; ISY1, Iridoid synthase; 7-DLS, 7-Deoxyloganetic acid synthase; 7-DLGT, 7-Deoxyloganetic acid glucosyltransferase; 7-DLH, 7-Deoxyloganic acid hydroxylase; LAMT, Loganic acid O-methyltransferase; SLS, Secologanin synthase; STR, Strictosidine synthase.

Figure 5 Protein-protein interaction network and gene expression profiling in the iridoid biosynthesis pathway of G. straminea.

(A) Protein-protein interaction network of the iridoid biosynthetic enzymes. Nodes represent pathways proteins, edges colors denote interaction evidence types: purple and light blue: Known interactions (database and experimental evidence): green: Gene neighborhood; red: Gene fusion; blue: Gene co-occurrence; yellow: Text mining; black: Coexpression; light purple: Protein homology. (B) Tissue-specific expression of iridoid-related isoforms. Heatmap displays row-wise z-score normalized FPKM values (red: high; blue: low). R, root; S, stem; L, leaf; F, flower; O, ovary; NEC, non-embryonic callus; EC, embryonic callus. AACT: Acetyl-CoA C-acetyltransferase; HMGCS: Hydroxymethylglutaryl-CoA synthase; HMGCR: Hydroxymethylglutaryl-CoA reductase (NADPH); MVK: Mevalonate kinase; PMK: Phosphomevalonate kinase; MVD: Diphosphomevalonate decarboxylase; IDI: Isopentenyl-diphosphate delta-isomerase; DXS: 1-Deoxy-D-xylulose-5-phosphate synthase; DXR: 1-Deoxy-D-xylulose-5-phosphate reductoisomerase; ISPE: 4-Diphosphocytidyl-2-C-methyl-D-erythritol kinase; GCPE: (E)-4-Hydroxy-3- methylbut-2-enyl-diphosphate synthase; ISPH: 4-Hydroxy-3-methylbut-2-en1yl diphosphate reductase; GPPS: Geranyl diphosphate synthase; GGPPS: Geranylgeranyl diphosphate synthase ; GGPPS SSU: Geranylgeranyl diphosphate synthase small subunits.

Bioinformatics analysis of G(G)PPS

Our results revealed that ten isoforms had GPPS/GGPPS annotations, three of which had open reading frames (ORFs). Two of these were annotated as GGPPS, while one isoform was annotated as GPPS, among the two GGPPSs, one was categorized as GGPPS small subunits (GGPPS SSU), and one was classified as a typical GGPPS. The prediction results revealed that the amino acid length of G(G)PPS (SSU) ranged from 342 to 424 aa, with corresponding molecular weights were 37.9 kDa and pI values were 5.81 (Table 1). Two of Gs GGPPS (SSU) possessed nagative GRAVY values, ranging from −0.050 to −0.187, indicating that these proteins have hydrophilicity. Gs GPPS had a positive GRAVY value (0.049), suggesting hydrophobicity of them. Three of Gs G(G)PPS (SSU) were identified no signal peptide. Two of Gs GGPPS (SSU) were localized in the chloroplast, Gs GPPS were predicted to be mitochondrion. Transmembrane structures were not predicted in any of the G(G)PPS proteins by TMHMM2.0 predictions (Table 2). Pfam protein structural domain prediction revealed a distinctive polyprenyl-synt domain shared by all the G(G)PPS proteins (Figs. S5A and S5B).

G(G)PPS usually contains two highly conserved aspartic acid-rich regions-FRAM and SARM with the sequences of DD(XX)1−2D (D is aspartic acid, and X refers to any amino acid). The first conserved region FRAM (DDXXXXD) is consistent with the binding site of the substrate dimethylallyl diphosphate (DMAPP), and the second conserved region SARM (DDXXD) corresponds to the binding site of the substrate isopentenyl diphosphate (IPP). which affects the catalytic activity of G(G)PPS. Some G(G)GPPS proteins also have the characteristic sequence CXXXC (C is cysteine, and X refers to any hydrophobic amino acid) of their structural domain, which is essential for proteins-proteins interactions (Beck et al., 2013). Sequence alignment results revealed that the Gs GGPPS SSU sequences were similar to those of Pj GGPPS SSU, Ae GGPPS SSU, Ca GGPPS SSU, and Si GGPPS SSU2, with identity values of 81.74%, 81.55%, 81.49% and 81.61%, respectively, according to DNAMAN (Table S7). The identities of the Gs GGPPS sequences were similar to those of Cr GGPPS, Ca GPPS, Ce GGPPS and Gj GGPPS, with values of 74.06%, 72.29%, 72.04% and 71.28%, respectively (Table S7). The Gs GPPS sequences were similar to those of Ce SPPS, Ca SPPS, Cr GPPS1, Cr GPPS2, Gsy FPPS, Si SPPS and Na SPPS, with identities of 91.84%, 91.76%, 92%, 91.84%, 91.53%, 90.68%, and 90.75%, respectively (Table S7). Gs GGPPS SSU were enriched with one FARM (DD(XX)2D) and two CXXXC regions. The Gs GGPPS subunit underwent a change in the second aspartic acid enrichment motif, from D to E, i.e, DDXXE (Fig. S5A). Gs GGPPS was enriched with one FARM region (DD(XX)2D), one SARM region each (DDXXD), and one CXXXC region (Fig. S5A). Gs GPPS was enriched with two SARM regions (DDXXD) (Fig. S5B).

Table 2 Physicochemical, structural properties and subcellular localization of GsG(G)PPS.

Isoform number	Gene name	length (aa)	MW (kD)	pI	SP	SL	GRAVY	TS	
Isoform0028091	GsGGPPS SSU	347	37.90001	5.81	NO	chloroplast	−0.187	o	
Isoform0030969	GsGGPPS	368	40.01208	6.28	NO	chloroplast	−0.050	o	
Isoform0027756	GsGPPS	424	46.45244	6.48	NO	mitochondrion	0.049	o	
Notes.

MW molecular weight

pI isoelectric point

SP Signal peptide cleavage site

SL Subcellular localization

GRAVY grand average of hydropathicity

TS Transmembrane structures

o indicates that the protein is predicted to be outside the membrane

The analysis of the conserved motifs in Gs G(G)PPS revealed that both Gs GGPPS SSU and Gs GGPPS contained motif 1, 2, 4, 6, 9. Gs GGPPS SSU and Gs GPPS shared motif 3, 5, 7. Gs GGPPS and Gs GPPS shared common motif 7 and 8 (Fig. 6). The difference in motif composition may influence the function of Gs G(G)PPSs, leading to changes in catalytic activity, protein subcellular localization, and other aspects. Protein structure prediction revealed that Gs GGPPS SSU, Gs GGPPS, and Gs GPPS exhibited high structural similarity to their homologous counterparts from Mucuna pruriens (GMQE = 0.85), Handroanthus impetiginosus (GMQE = 0.81), and Catharanthus roseus (GMQE = 0.79), respectively. Gs G(G)PPS mainly contained α-helices and random coils in the tertiary structure (Fig. S6).

Figure 6 Conserved motif analysis of G(G)PPS in G. straminea.

Phylogentic analysis of G(G)PPSs

Phylogenetic analysis revealed that the G(G)PPSs identified can be categorized into three distinct branches. Among them, Gs GGPPS, together with large subunits of GGPPS (GGPPS LSUs) and GGPPSs from other species, clustered into group 1. Gs GGPPS SSU along with the small subunits of GGPPS (GGPPS SSUs) from other species, was grouped into the second branch (group 2). Gs GPPSs formed the third branch, together with GPPS, SPPS, and FPPS from various other species (group 3) (Fig. 7). Gs G(G)PPSs were categorized into three distinct groups based on their sequence and functional divergence.

Figure 7 Phylogenetic tree of the G(G)PPS gene family across different species.

The abbreviations and sequence ID of G(G)PPS gene family are shown in Table S1. Red triangles, red circles and red stars indicated the proteins annotated in this study. the numbers on the branches represent the statistical support values for those branches. These values are obtained through bootstrap analysis, with higher numbers indicating more reliable and credible branch divisions.

Expression analysis by real-time quantitative PCR

To further analyze the expression patterns of genes annotated to iridoid biosynthesis pathway in different tissues, RT-qPCR was performed on key genes of the iridoid biosynthesis pathway. We selected the upstream and downstream initiators of the MVA (AACT, MVD) and MEP (DXS, ISPH, GCPE) pathways, and these genes (IDI, GPPS) serving as key genes involved in intermediate production and skeletal formation steps. As shown in Fig. 8, the qPCR analysis revealed distinct tissue-specific expression patterns of these genes; AACT, IDI, ISPH, and GCPE showed their highest expression levels in leaves, DXS and GPPS were most abundantly expressed in stems, while MVD exhibited peak expression in non-embryogenic callus. Conversely, the lowest expression levels were observed for AACT, DXS, and ISPH in root tissues, and for IDI, MVD, GCPE, and GPPS in flowers. Significant differential expression across tissues was observed for all genes, with particularly pronounced variations for ISPH, GCPE, and GPPS. The RNA-seq results demonstrated some similar but also distinct patterns: AACT and GPPS showed highest expression in embryogenic callus, ISPH and GCPE in leaves, while MVD was uniquely most highly expressed in roots. For most genes (DXS, IDI, ISPH, GCPE, GPPS), the expression trends across different tissues were generally consistent between qPCR and RNA-seq data. However, discrepancies were noted for AACT and MVD in roots and leaves, as well as for DXS and MVD in the ovary.

Figure 8 Tissue-specific expression abundances of key genes involved in iridoid synthesis.

Note: Bar chart indicated the relative expression levels of genes, line chart indicated the FPKM values of genes. NEC indicated non-embryonic callus, EC indicated embryonic callus. Bars represent standard deviation; Different lowercase letters indicating significant differences at the 0.05 level of probability according to Duncan’s multiple-rangetest.

Discussion

As an important medicinal plant, G. straminea produces diverse iridoids, its principal bioactive constituents, which are primarily biosynthesized via the terpenoid pathway. Transcriptome analysis revealed significant enrichment of genes associated with secondary metabolite synthesis, especially terpenoid backbone biosynthesis, with 121 genes annotated to this pathway. This genetic repertoire presumably underlies the molecular foundation for iridoid production in the plant. Comparing with the results obtained for G. straminea via Illumina NGS (Zhou et al., 2016), and Gentiana waltonii and Gentiana robusta via the Illumina Hiseq X Ten platform (Ni et al., 2019), we obtained more annotation information than previous studies, significantly enriching the genomic resources for G. straminea, and providing a more comprehensive genetic database.

TFs can regulate gene expression by recognizing specific DNA sequences in gene promoters, which is important for understanding gene expression regulatory mechanisms (Jose, Franco & Roberto, 2016). In plants, major TF families including GRAS, bHLH, and WRKY are implicated in hormone signaling and secondary metabolism. Our annotation identified predominant TF distributions in the GRAS, ARF, C3H, bHLH, WRKY, and FAR1 families, consistent with their metabolic regulatory functions. As members of the GRAS family, the SmDELLA1 protein positively regulates phenolic acid and flavonoid biosynthesis (Li et al., 2024). DELLA proteins additionally modulate jasmonic acid (JA) signaling and cell wall formation (Hou et al., 2010; Wang et al., 2021). As the second largest angiosperms TF class, bHLHs govern epidermal differentiation, stress responses, and secondary metabolism. They are key regulators of anthocyanin biosynthesis (Jaakola, 2013) and loganic acid production (Fu et al., 2024). Plant-specific WRKYs bind conserved W-box elements to activate downstream genes (Brand et al., 2013). Artemisia carvifolia AaWRKY1 promoted artemisinin biosynthesis via activating sesquiterpene synthase expression (Ma et al., 2009).

Iridoids are present in traditional medicinal plants and regulate various diseases in the human body. The synthesis of iridoids has been reported in C. roseus (Oudin et al., 2007), Gentiana rigescens (Zhang et al., 2015), Valeriana jatamansi (Zhao and Wang., 2020), Swertia mussotii (Liu et al., 2017) and Rehmannia glutinosa (Sun et al., 2012). In our study, 117 isoforms involved in 19 key enzymes were annotated, spanning multiple stages of iridoid synthesis. Comparative analyses reveal species-specific patterns: for instance. S. mussotii exhibited 39 transcripts link to 24 enzyme categories (Liu et al., 2017), Gentiana lhassica had 171 unigenes encoding 27 key enzymes (Heng et al., 2021), and in V. jatamansi contained 24 unigenes associated with three metabolic pathways (Zhao & Wang, 2020). In Panax ginseng (Kim et al., 2014) and Ganoderma lucidum (Shi et al., 2012), overexpressed MVD could significantly increase the accumulation of terpenoids in plants. The overexpression of HDR gene in Artemisia annua (Ma et al., 2017) and Ginkgo biloba (Kim et al., 2021) could significantly increase the terpenoids content.

Studies show that genes typically do not act alone but co-express with related genes in their metabolic pathways to achieve specific functions (Zhao et al., 2013a; Zhao et al., 2013b). Some research has demonstrated that co-expression of G(G)PPS and GES genes significantly enhances the accumulation of terpenoid indole alkaloids (TIAs) (Kumar et al., 2015). Upregulation of DXS expression markedly increases the expression of its downstream gene GGPPS, thereby promoting carotenoid biosynthesis (Henriquez et al., 2016). Co-expression of DXS and GGPPS in Salvia miltiorrhiza resulted in transgenic hairy roots with substantially higher tanshinone content compared to single-gene transformants and wild-type controls (Shi et al., 2016). Overexpression of LcGPPS.SSU1 enhanced DXS expression level in the metabolic pathway via a positive feedback regulatory mechanism, concomitantly elevating GGPPSs expression levels (Zhao et al., 2020). Elevated ISPG ((E)-4-hydroxy-3-methylbut-2-enyl diphosphate synthase, also designated GCPE) levels induced HMBPP accumulation, which subsequently activated ISPH to alleviate both HMBPP buildup and the negative effects of GCPE overexpression (Li et al., 2017). Coordinated expression of GCPE and ISPH simultaneously eliminated the accumulation of both HMBPP and MECPP, leading to a dramatic increase in isoprenoid production (Li et al., 2017). Deficiency in ISPE activity reduces the levels of MEP pathway metabolites (chlorophylls and carotenoids), while stimulating the expression of ISPF and ISPG (Ahn & Pai, 2008). Methylerythritol cyclodiphosphate (MEcPP) may regulate MEP pathway activity through feedback modulation of DXS protein abundance (Wang et al., 2019). PtHMGR overexpression resulted in transcriptional activation of genes across the MEP and MVA pathways (including DXS, IDI, and GPS), suggesting a global regulatory role (Wei et al., 2019). WRKY transcription factors likely participate in terpenoid biosynthesis by binding to W-box elements (TTGACC) in the promoters of both MEP pathway genes (IDS, DXS) and MVA pathway genes (HMGS) (Meng et al., 2018). When ISPC (1-Deoxy-D-xylulose-5-phosphate synthase, also known DXR), ISPE, ISPH and ISPG were expressed as a tandem gene cluster under a shared promoter in the host strain, taxadiene production was significantly improved (Huang & Zhong, 2017).

These regulatory mechanisms ultimately converge to optimize the production of terpenoid precursors. GPP/GGPP serve as essential terpenoid precursors synthesized from IPP and DMAPP through distinct enzymatic pathways. Researchers have demonstrated that GPPS catalyzes the condensation of IPP and DMAPP form GPP for monoterpene synthesis, while mediates their conversion to GGPP as the precursor for diterpene synthesis, triterpene synthesis, etc. (Tholl et al., 2004; Liang, Ko & Wang, 2002). Since both of them act on the same substrate, some scholars have hypothesized that the IPP flow direction determines the product differences (Tholl et al., 2004). In the third stage, geraniol is formed via the action of GES, and then 10-hydroxygeraniol is formed via the catalytic action of G10H (Liang, Ko & Wang, 2002). The genes encoding G10H in C. roseus (Krithika et al., 2015) and S. mussotii (Wang et al., 2010) have been cloned. Although the G10H gene was not annotated in our results, we detected its transfer partner cytochrome P450 reductase (CPR, POR, EC1.6.2.4); this was the partner of G10H, in the catalytic production of 10-hydroxygeraniol from geraniol. Previous studied have demonstrated that cytochrome P450 monooxygenases (e.g., CYP76B6 in C. roseus and CYP76B10 in S. mussotii) can catalyze the conversion of geraniol to loganic acid (Wang et al., 2010; Collu et al., 2001; Hofer et al., 2013). Notably, the catalytic activity of plant CYP450s depends on electron transfer mediated by POR (Durst & Nelson, 1995). And their expression patterns were similar to G10H (Hofer et al., 2013). Base on relevant reports in R. glutinosa (Sun et al., 2012), the POR annotated in this study may catalyze geraniol formation.

These cytochromes P450-mediated conversions represent critical downstream modifications of terpenoid precursors. At the upstream level, the biosynthesis of these precursors is tightly regulated by prenyltransferases. GPP synthase, a member of the short chain prenyltransferase family, catalyzes the condensation of DMAPP and IPP to form GPP. Both FPPS and GGPPS belong to this enzyme family. These prenyltransferases not only regulate IPP flux but also exhibit conserved structural characteristics across plant species (Durst & Nelson, 1995). Our result revealed that the amino acid sizes, molecular weights and isoelectric points of Gs G(G)PPS annotated in this study were essentially similar to those reported for GGPPS in S. miltiorrhiza (Li et al., 2024), Liriodendron tulipifera (Zhang et al., 2021) and wintersweet flower (Kamran et al., 2020). The characteristic conserved motif of Gs GGPPS SSU was consistent with that of Cp GPPS.SSU2 and Cp GPPS.SSU1 reported in wintersweet flower (Kamran et al., 2020). The Gs GGPPS was consistent with the Ltu GGPPS2 reported in the Liriodendron tulipifera (Zhang et al., 2021) and the Cp GPPS reported in wintersweet flower (Kamran et al., 2020). The characteristic conserved motif of Gs GPPS was similar to other characteristics of homologous GPPSs (Kamran et al., 2020).

G(G)PPS was shown to exist in both homologous and heterologous forms in the plant material (Chen, Fan & Wang, 2015), heterodimeric G(G)PPS contained one LSU and one SSU, and the LSU of the heterodimeric GPPS showed 50%–75% sequence similarity to that of GGPPS and possessed isopentenyl transferase activity, which catalyzes the production of mainly GGPP, as well as a small amount of GPP and FPP (Tholl et al., 2004; Kamran et al., 2020). However, the heterodimeric GPPS SSU shares little sequence similarity to with GGPPS, only 22%–38%, lacks the DD(XX)1−2D motif, and shows no isoprenyl transferase activity (Tholl et al., 2004). Five full-length GPPS and GGPPS genes were successfully annotated in the wintersweet flower transcriptome, these genes were classified into three branches by phylogenetic analysis, namely the SSU representing the heterodimeric GPPS and the homodimeric GPPS and GGPPS (Kamran et al., 2020).

Studies demonstrated that GGPPS large subunit (LSU) can form heterodimeric with GPPS inactive small subunit (SSU) to catalyze monoterpene precursor substances. For instance, homologous and heterologous GPP synthetases have been identified in C. roseus, and classified as the LSU of Cr GPPS, the SSU of heterologous Cr GPPS, and homologous Cr GPPS, the LSU of Cr GPPS is bifunctional in the formation of GPP and GGPP, whereas the inactive SSU of Cr GPPS can integrate with Cr GPPS LSU, increasing enzyme activity, and result in the production of only GPP (Rai et al., 2013). It was hypothesized that the inactive SSU of the heterodimeric Cr GPPS interacting with the bifunctional G(G)PPS redirected metabolic flux towards, and thus acting as an important regulator of monoterpene indole alkaloid biosynthesis (Zhang et al., 2021). Similar mechanisms occured in Arabidopsis (Orlova et al., 2010), Antirrhinum majus (Tholl et al., 2004) and C. roseus (Zhang et al., 2021). In tobacco, the studies further shown that Am SSU overexpression enhanced GPPS activity in leaves and flowers and promoted monoterpene production (Orlova et al., 2010). These findings collectively indicate that both homodimeric and heterodimeric G(G)PPS are regulated monoterpenes formation across species, with LSU either binding SSU or functioning as a homodimer to regulate the flow of IPPs. However, the reason for this phenomenon in G. straminea remains still unclear, and requires further investigation.

RNA-seq covers nearly all exon regions of a gene, so the obtained gene expression levels effectively represent the combined expression across all exonic regions. In contrast, qPCR quantifies expression by amplifying a localized region using designed primers and does not account for the full-length gene (Everaert et al., 2017). Furthermore, RNA integrity and purity, cDNA synthesis efficiency, and PCR amplification efficiency may influence the qPCR results (Lin et al., 2023). Additional research has indicated that GC content also has a strong sample-specific impact on gene expression results (Hansen, Irizarry & WU, 2012; Love, Hogenesch & Irizarry, 2016). For some low-expression genes, expression estimates may be less accurate (Li, Jiang & Wong, 2010). In this study, real-time qPCR validation demonstrated strong concordance between gene expression profiles and transcriptome sequencing data for the majority of analyzed genes. However, certain genes exhibited inter-platform discrepancies, a phenomenon consistent with established methodological variance reports. The overall high correlation (85%) between qPCR and RNA-seq data aligns with previously reported plantform concordance metrics (Everaert et al., 2017). However, the observed inter-platform discrepancies for certain genes could be attributed to the complex, multi-layered control system governing plant secondary metabolism, which includes transcriptional, post-transcriptional, translational, and post-translational processes (Hemmerlin, Harwood & Bach, 2012; Deng et al., 2024; Zhao et al., 2023). Our study integrated RT-qPCR and RNA-seq data to reveal significantly higher expression levels of representative iridoid biosynthesis-related genes in aerial tissues. These findings were corroborated by previous studies: Rai demonstated in C. roseus that GPPS exhibited peaked in the flower tissues, followed by the stem, across examined organs (roots, stems, leaves, flowers, and siliques) (Rai et al., 2013). Similarly, Zhou et al. (2016) reported that GPPS exhibited higher expression levels in the flowers comparing to root. Collectively, these results highlight the species-specific tissue distribution patterns of genes involved in iridoid biosynthesis pathways.

In traditional Chinese medicine, the root of Gentiana spp. serve as the official medicinal parts, while Tibetan medicine also utilizes aerial portions (stems, leaves, and flowers) (Yang, 1991). Some studies also have reported superior medicinal quality in the aerial parts of Gentiana species (Zhao, Dorje & Wang, 2010). Similar results were also observed in transcriptome analyses of G. crassicaulis, G. lhassica and G. Rhodantha (Kang et al., 2021a; Kang et al., 2021b; Zhang et al., 2023). Moreover, quantitative analysis of gentiopicroside and loganic acid content across different tissues (roots, stems, leaves, and flowers) of G. lhassica demonstrated markedly higher total iridoid accumulation in aerial organs than in underground parts, and with statistically significant differences (Kang et al., 2021b; Zhang et al., 2023). This suggests that they may possess higher quality than the underground parts.

In summary, this study delineates the tissue-specific expression landscape of iridoid biosynthesis genes in G. straminea through transcriptome analysis, with aerial-part-enriched key genes revealing molecular insights into its medicinal compound accumulation. This work fills critical knowledge gaps underlying iridoid biosynthesis in this species and provides valuable candidate targets (through the annotation of key genes) for future metabolic engineering optimization of iridoid compounds and heterologous biosynthesis of medicinal components.

Conclusions

Combining Pac-Bio long-read and Illumina short-read (HiSeqTM 4000) sequencing, this study conducted full-length transcriptome assembly and differential expression analysis across five tissues of G. straminea. A total of 32,776 full-length transcripts of high-quality without redundancy were obtained, and 31,434 isoforms were annotated in the NR, KEGG, KOG and Swiss-Prot databases. Illumina sequencing revealed 31,330 genes that were commonly expressed across all five tissues. According to KEGG enrichment analysis, the DEGs were mainly enriched in biosynthesis of secondary metabolites, metabolic pathways, mitogen-activated protein kinase (MAPK) signaling pathway. In summary, 708 genes were classified into 20 KEGG secondary metabolism pathways in the transcriptome of G. straminea. All genes involved in the biosynthesis of iridoids were screened, and a total of 117 isoforms were annotated into the iridoid synthesis pathway, resulting in the identification of key genes encoding 19 enzymes. RT-qPCR results showed that AACT, IDI, ISPH, and GCPE had the highest expression levels in leaves, while DXS and GPPS had the highest expression levels in stems. DXS, IDI, MVD, ISPH, and GPPS exhibited the higher expression levels in NEC than in EC, RT-qPCR results showed a similar trend in expression abundance across the tested tissues. The polyprenyl_synt domain was highly conserved in both the identified GsGGPPSs and GsGPPSs. Through phylogenetic analysis, the GsG(G)PPSs annotated in this study could be classified into three branches. These new results provide valuable information for further research on functional gene development and active ingredient accumulation patterns in G. straminea.

Supplemental Information

Supplemental Information 1 Sequence information of proteins analyzed in this study

Supplemental Information 2 Primer for real-time quantitative PCR

Supplemental Information 3 Gene function annotation via KEGG metabolic pathway classification

Supplemental Information 4 The ten pathways with the highest number of annotated genes in the KEGG

Supplemental Information 5 The number of genes involved in secondary metabolism according to the KEGG pathway analysis

Supplemental Information 6 Isoforms involved in iridoid biosynthesis

“ –” i ndicatesnoE . C . number or KEGG Orthology ( KO )

Supplemental Information 7 Protein sequence homology of GsG(G)PPS

Note: in G. straminea , GsGGPPS SSU denotes the small subunits of geranylgeranyl diphosphate synthase ; GsGGPPS represents the geranylgeranyl diphosphate synthase; GsGPPS indicates geranyl pyrophosphate synthase .

Supplemental Information 8 SMRT sequencing of G.straminea

(a) CCS read length distribution; (b) CCS pass distribution; (c) Consensus isoform distribution; (d) Length distribution of transcript isoforms .

Supplemental Information 9 GO function classification

Supplemental Information 10 Gene number of each TF family (top ten)

Supplemental Information 11 KEGG pathway enrichment of DEGs in different groups

Supplemental Information 12 Multiple sequence alignment of deduced amino acid sequences of GGPPSSSU, GGPPS, GPPS-like proteins from G. straminea and other plant species

(a) Sequence alignment of GsGGPPS(SSU) with GGPPS(SSU) from other plants; (b) Sequence alignment of GsGPPS with GPPS, SPPS from other plants; Identical residues are shaded in dark blue, highly similar residues are shaded in pink, and similar residues are shaded in light blue. Polyprenyl-synt domain was marked with purple border; DDXXXXD FARM and DDXXD (SARM) motif was marked with red border, CxxxC motif was marked with green border; Aspartate-rich region was marked with red Triangle; Chain length determination region was marked with green horizontal line; Active site lid residues was marked with blue horizontal line. Gene abbreviations, the species and gene ID information were shown in Table S1.

Supplemental Information 13 Protein tertiary structure of G(G)PPS of G . straminea

Structure prediction was performed by homologous modelling with the SWISS-MODEL sever, GsGGPPS SSU using structure of the geranylgeranyl pyrophosphate synthase small subunit protein of Mucuna pruriens (velvet bean) (A0A371F419) as the template. GsGGPPS using structure of the geranylgeranyl pyrophosphate synthase protein of Handroanthus impetiginosus GGPPS (A0A2G9GV50) as the template, GsGPPS using structure of the geranyl pyrophosphate synthase protein of Catharanthus roseus (B2MV87) as the template. (a) Predicted structure of GsGGPPS SSU shown; (b) Predicted structure of GsGGPPS shown; (c) Predicted structure of GsGPPS shown.

Supplemental Information 14 The detailed procedure for the PCR experiment

Supplemental Information 15 MIQE checklist

Supplemental Information 16 The average Ct of q-PCR

Supplemental Information 17 Gs Transcriptome Functional annotation

Supplemental Information 18 Gs Isoforms sequence

The authors would like to thank the anonymous reviewers for their valuable comments and suggestions.

Additional Information and Declarations

Competing Interests

Author Contributions

Data Availability

The authors declare that they have no conflicts of interest.

Lina Yang conceived and designed the experiments, performed the experiments, analyzed the data, prepared figures and/or tables, authored or reviewed drafts of the article, and approved the final draft.

Tao He conceived and designed the experiments, authored or reviewed drafts of the article, and approved the final draft.

Le Wang conceived and designed the experiments, analyzed the data, prepared figures and/or tables, authored or reviewed drafts of the article, and approved the final draft.

Xiaochun Ning performed the experiments, prepared figures and/or tables, sample collection, and approved the final draft.

Shuai Wang performed the experiments, analyzed the data, prepared figures and/or tables, and approved the final draft.

The following information was supplied regarding data availability:

The data is available at China National Center for Bioinformation: CRA017932, CRA019968.

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
