# Peer review of "Full-length transcriptome profiling of Gentiana straminea Maxim. provides new insights into iridoid biosynthesis pathway"

_PeerJ, doi:10.7717/peerj.20136_

## Round 0.1 · original submission · Major Revisions

The manuscript was reviewed by four independent experts in the field. All reviewers found the work interesting but raised several issues which should be addressed for further consideration. The reviewers provide detailed comments in their reviews and point out the areas where the manuscript needs to be improved.

**Language Note:** The review process has identified that the English language must be improved. PeerJ can provide language editing services - please contact us at [email protected] for pricing (be sure to provide your manuscript number and title). Alternatively, you should make your own arrangements to improve the language quality and provide details in your response letter. – PeerJ Staff

Reviewer 1 ·

Basic reporting

overall the research article has reported many important aspects to the important traditional chinese medicine and its important secondary metabolite of iridoid class. Some minor edits in the phrasing of some sentence grammar can be important.
some of the examples are Lines 23, 77, 121, 128.

Experimental design

no comments.

Validity of the findings

1. Particular reporting of the GPPS enzymes that are located at different parts of the cell. can the authors answer the question of whether there are different classes of enzymes specific for cell organelle . and if they are similar; what makes there function important to the specific location.

2. the gene expression of key enzymes at different parts of the plant: is there any sporting study authors should mention that proves the yields of secondary metabolites production is affected but the expression of such enzymes?

3. PPI networks the authors generated: is there any possibility to generated co-expression based of expression of enzymes, supporting the PPI network conclusions?

·

Basic reporting

The authors described a detailed background and designed the experimental procedures with clear objectives.

There are minor grammatical mistakes throughout the text which can be improved.

The resolution and aspect ratios of the figures 3 & 4 need attention.

Experimental design

No comment

Validity of the findings

Page 18 Line 384.
Did the authors consider verifying this hypothesis by performing quantitation of iridoid metabolites?
A quantitative analysis (e.g. LCMS) of iridoid metabolites in different tissues would further strengthen the findings of this study.

Additional comments

The authors of this study performed transcriptomic profiling of different tissues in a medicinally relevant G. straminea plant. Differential gene expression analysis revealed a putative pathway for iridoid biosynthesis, a bioactive compound, and key proteins involved in the process. This study provides a comprehensive overview of functional genes and associated metabolic pathways with an emphasis on the biosynthesis of iridoid compounds.

·

Basic reporting

The authors have conducted good piece of work on iridoid biosynthesis of G.straminea.

Experimental design

Satisfactory

Validity of the findings

Satisfactory

Additional comments

No Comments

Reviewer 4 ·

Basic reporting

Please see additional comments.

Experimental design

Please see additional comments.

Validity of the findings

Please see additional comments.

Additional comments

In this manuscript, Yang et al., performed transcriptome profiling and assembly to study genes involved in iridoid biosynthesis in the medicinal plant Gentiana straminea. While the dataset could be valuable and merit publication, the execution of visualizations was poor.

Major comments:
1. In the abstract it is unclear whether Illumina or PacBio sequencing technologies are used. The phrase “Full-length transcriptome-based Illumina” is confusing, since “full-length” would imply the data were generated by PacBio. In the main text it is described that the author used a combined sequencing technology, which should be made clearer and more explicit in the abstract.

2. Through out the manuscript, some acronyms are not defined in their first mention. For example, in line 130, CCS is not defined.

3. In lines 136-137, by “consistent sequence”, do you mean consensus sequence?

4. Most figures are pixelated and almost illegible. This might be due to the submission system. The main text figures also seemed “squeezed” in the horizontal direction.

5. For figure 2, is the x axis label supposed to be in alphabetical order? If not, what is the basis for the ordering?

6. The text of figure 4 is too small and illegible. When I zoom in, it is too pixelated.

7. In the introduction, the author stated that the roots of G. straminea has been used as herbal medicine. Can the authors discuss if the iridoid biosynthetic genes should be most highly expressed in the root?

8. How was the PPI analysis performed? There was very little details in the Methods for this analysis. Did the authors mean gene co-expression analysis? What are the full names of the proteins in the network? Has any of the predicted interaction been validated or supported by the literature? If not, this analysis should be removed.

9. Lines 319-320: 6 out of the 10 GPPS/GGPPS isoforms have ORFs. Why would a transcript without ORF be kept in downstream analyses? Of all isoforms, how many have ORFs? If a transcript does not have a ORF, maybe they are truncated (missing start and/or stop codons).

10. For Figure 8, how was phylogeny built? There was very little detail in the Methods for this figure.

11. For Figure 9, why were these genes chosen? And why were other genes, such as ISY, not chosen?

---

## Round 0.2 · Minor Revisions

Please note that the Section Editor has some concerns that need to be addressed:

- Citations are not in alphabetical order (example Ozsolak is listed before Orlova; there are others.). It appears that there were only sorted by first letter of the last name. The authors should consider using a citation manager such as Zotero (free!)

- A fasta file of the final isoforms should be deposited into a public repository (ideal) or at least provided as supplemental data

- annotation of the final isoforms should be provided

- (minor): sometimes "DGEs" is used when I think the authors mean "DEGs"

- (lines 156-157) "The results were expressed in terms of FPKM. Differential analysis of gene
expression in different tissues was performed by using DESeq2 software". This is incorrect methodology. DEseq2 requires input of raw read counts NOT FPKM.

- (line 318) "By combining these results with previous research results we identified a putative pathway for iridoid biosynthesis ". More detail about the methodology for identification of the iridoid biosynthesis pathway needs to be given. There is nothing in the methods about this and it is the crux of the paper.

---

## Round 0.3 · accepted · Accept

Thank you for addressing the remaining concerns. The revised version is acceptable now.